# Contextual Recurrent Convolutional Model for Robust Visual Learning

## Abstract

Feedforward convolutional neural network has achieved a great success in many computer vision tasks. While it validly imitates the hierarchical structure of biological visual system, it still lacks one essential architectural feature: contextual recurrent connections with feedback, which widely exists in biological visual system. In this work, we designed a Contextual Recurrent Convolutional Network with this feature embedded in a standard CNN structure. We found that such feedback connections could enable lower layers to "rethink" about their representations given the top-down contextual information. We carefully studied the components of this network, and showed its robustness and superiority over feedforward baselines in such tasks as noise image classification, partially occluded object recognition and fine-grained image classification. We believed this work could be an important step to help bridge the gap between computer vision models and real biological visual system.

## 1 Introduction

It has been long established that the primate's ventral visual system has a hierarchical structure (Felleman & Van Essen, 1991) including early (V1, V2), intermediate (V4), and higher (IT) visual areas. Modern deep convolutional neural networks (CNNs) for image recognition (Krizhevsky et al., 2012; Simonyan & Zisserman, 2014) trained on large image data sets like ImageNet (Russakovsky et al., 2015) imitate this hierarchical structure with multiple layers. There is a hierarchical correspondence between internal feature representations of a deep CNN's different layers and neural representations of different visual areas (Cichy et al., 2016; Yamins & DiCarlo, 2016); lower visual areas (V1, V2) are best explained by a deep CNN's internal representations from lower layers (Cadena et al., 2017; Khaligh-Razavi & Kriegeskorte, 2014) and higher areas (IT, V4) are best explained by its higher layers (Khaligh-Razavi & Kriegeskorte, 2014; Yamins et al., 2014). Deep CNNs explain neuron responses in ventral visual system better than any other model class (Yamins & DiCarlo, 2016; Kriegeskorte, 2015), and this success indicates that deep CNNs share some similarities with the ventral visual system, in terms of architecture and internal feature representations (Yamins & DiCarlo, 2016).

However, there is one key structural component that is missing in the standard feedforward deep CNNs: contextual feedback recurrent connections between neurons in different areas (Felleman & Van Essen, 1991). These connections greatly contribute to the complexity of the visual system, and may be essential for the success of the visual systems in reality; for example, there are evidences that recurrent connections are crucial for object recognition under noise, clutter, and occlusion (O'Reilly et al., 2013; Spoerer et al., 2017; Rajaei et al., 2018).

In this paper, we explored a variety of model with different recurrent architectures, contextual modules, and information flows to understand the computational advantages of feedback circuits. We are interested in understanding what and how top-down and bottom-up contextual information can be combined to improve in performance in visual tasks. We investigated VGG16 (Simonyan & Zisserman, 2014), a standard CNN that coarsely approximate the ventral visual hierarchical stream, and its recurrent variants for comparison. To introduce feedback recurrent connections, we divided VGG16's layers into stages and selectively added feedback connections from the groups' highest layers to their lowest layers. At the end of each feedback connection, there is a contextual module (Section 3.2) that refines the bottom-up input with gated contextual information. We tested

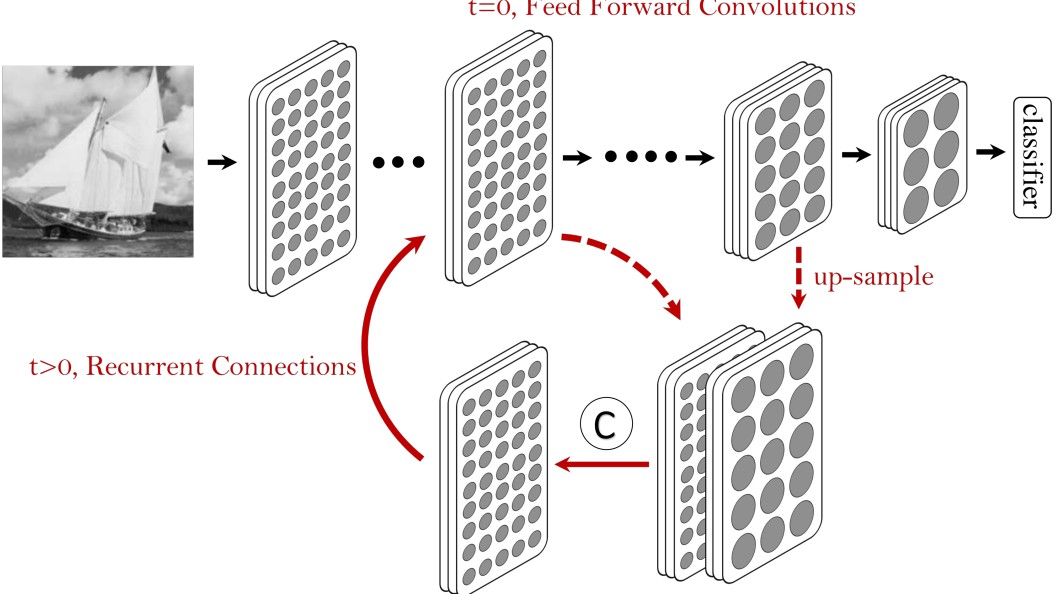

Figure 1: The schematic of a Contextual Recurrent Convolutional Network (CRCN). Check Section 3.1 for details.

and compared several networks with such contextual modules against VGG16 in several standard image classification task, as well as visual tasks in which refinement under feedback guidance is more likely to produce some beneficial effects, such as object recognition under degraded conditions (noise, clutter and occlusion) and fine-grained recognition. We found that our network could outperform all the baseline feedforward networks and surpassed them by a large margin in fine-grained and occlusion tasks. We also studied the internal feature representations of our network to illustrate the effectiveness of the structure. While much future work has to be done, our work can still be an important step to bridge the gap between biological visual systems and state-of-the-art computer vision models.

## 2    RELATED WORK

Although recurrent network modules including LSTM (Hochreiter & Schmidhuber, 1997) and Gated Recurrent Unit (Cho et al., 2014) have been widely used in temporal prediction (Wang et al., 2017c) and processing of sequential data (e.g. video classification (Donahue et al., 2015)), few studies have been done to augment feedforward CNNs with recurrent connections in image-based computer vision tasks.

**Image classification.**    Standard deep CNNs for image classification suffer from occlusion and noise (Wang et al., 2017a;b; Zhang et al., 2017), since heavy occlusion and noise severely corrupt feature representations at lower layers and therefore cause degradation of higher semantic layers. With the inclusion of feedback connections, a model can "rethink" or refine its feature representations at lower layers using feedback information from higher layers (Li et al., 2018); after multiple rounds of feedback and refinement, input signals from distracting objects (noise, irrelevant objects, etc.) will be suppressed in the final feature representation (Cao et al., 2015). Li et al. (2018) used the output posterior possibilities of a CNN to refine its intermediate feature maps; however, their method requires posterior possibilities for refinement and thus cannot be applied in scenarios where supervision is absent. Jetley et al. (2018) used more global and semantic features at higher convolutional layers to sharpen more local feature maps at lower layers for image classification on CIFAR datasets; however, our own experimentation suggests that this method only works when the higher and lower layers have a relatively small semantic gap (similarly sized receptive fields); on high-

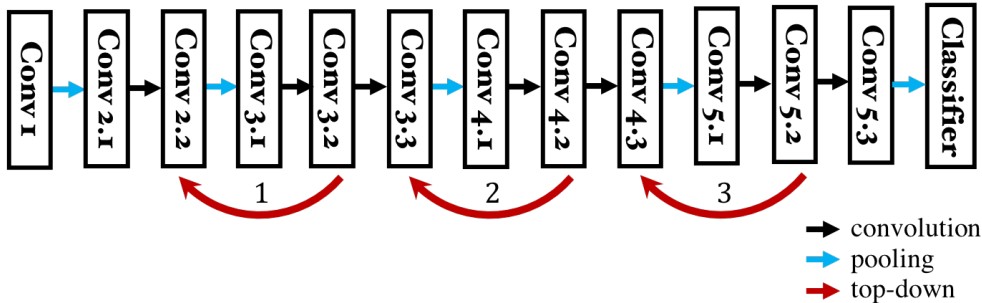

Figure 2: The details of a VGG-style context-gating recurrent model.

resolution dataset like ImageNet, large semantic gaps between higher and lower layers make this method difficult to work.

**Other computer vision tasks.** Linsley et al. (2018) designed a model with explicit horizontal recurrent connections to solve contour detection problems, and Spoerer et al. (2017) evaluated the performance of various models with recurrent connections on digit recognition tasks under clutter. The tasks evaluated in these studies are rather simple and contrived, and it remains to be seen whether their models and conclusions can apply to real world computer vision problems. (Li et al., 2018) uses posterior possibilities at the last fully connected layer to select intermediate feature map representations; however, the posterior possibility vector is not informative enough and the input of the feedback connection is totally fixed, which makes it less flexible to fully mimic the recurrent connections in the visual system. Overall, feedback and recurrent connections are present in multiple layers of the visual hierarchy, and this study constrains feedback connections to the output classification layer only. It is worth noting that a recent study (Nayebi et al., 2018) is motivated by recurrent connections in the brain as well; however, their work focuses on exploring the computational benefits of local recurrent connections while ours focuses on feedback recurrent ones. Thus, we believe that our work is complementary to theirs.

## 3 METHODS

In this section, we will describe the overall architecture of our proposed model and discuss some design details.

### 3.1 OVERALL MODEL ARCHITECTURE

The main structure of our Contextual Recurrent Convolutional Network (CRCN) is shown in Figure 1. A CRCN model is a standard feedforward convolutional network augmented with feedback connections attached to some layers. At the end of each feedback connection, a contextual module fuses top-down and bottom-up information (dashed red lines in Figure 1) to provide refined and sharpened input to the augmented layer.

Given an input image, the model generates intermediate feature representations and output responses in multiple time steps. At the first time step ($t = 0$ in Figure 1), the model passes the input through the feedforward route (black arrows in Figure 1) as in a standard CNN. At later time steps ($t > 0$ in Figure 1), each contextual module fuses output representations of lower and higher layers at the previous step (dashed red lines in Figure 1) to generate the refined input at the current time step (red lines in Figure 1). Mathematically, we have

$$O_k^{(t)} = \begin{cases} f_k(O_{k-1}^{(t)}) & \text{if } t = 0 \text{ or } k \notin S_G \\ c_k(O_{k-1}^{(t-1)}, O_{h(k)}^{(t-1)}) & \text{if } t > 0 \text{ and } k \in S_G \end{cases}, \quad (1)$$

where $S_G$ is the index set of layers augmented with feedback connections and contextual modules, $c_k(\cdot, \cdot)$ (detailed in Eqs. (2)) is the contextual module for layer $k$, $O_k^{(t)}$ denotes the output of layer $k$ at time $t$, $h(\cdot)$ is a function that maps the index of an augmented layer to that of its higher feedback

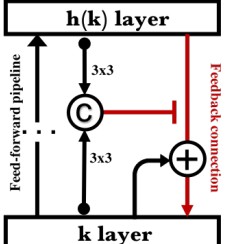
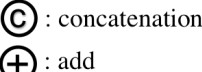

Ⓒ : concatenation

⊕ : add

⊣⊢ : dot product

Figure 3: The schematic of our proposed contextual module. Layer $k$ denotes the bottom-up layer and layer $h(k)$ denotes the top-down layer aligned with the size of k layer. The left black arrow shows the feed-forward pipeline.

| Model | CIFAR-10 | CIFAR-100 |
|---|---|---|
| VGG-small | 91.20 | 67.06 |
| VGG-ATT | 91.77 | 69.48 |
| VGG-LR-2 | 91.49 | 68.99 |
| VGG-CRCN-1 | **92.37** | 70.82 |
| VGG-CRCN-2 | 91.88 | **71.55** |

Table 1: Top-1 image classification accuracy on CIFAR datasets. VGG-small means VGG model with only one FC layer. VGG-ATT means the model proposed in (Jetley et al., 2018), VGG-LR-2 means the "rethinking" one-FC-layer VGG model with 2 unrolling times proposed in (Li et al., 2018). CRCN-*n* means our 2-recurrent-connection model with *n* unrolling times.

layer, and $f_k(\cdot)$ denotes the (feedforward) operation to compute the output of layer $k$ given some input.

## 3.2 CONTEXTUAL MODULE DESIGN

The key part of the Contextual Recurrent Convolutional Network model is the contextual module at the end of each feedback connection. Figure 3 shows one possible design of the contextual module, which is inspired by traditional RNN modules including LSTM (Hochreiter & Schmidhuber, 1997) and Gated Recurrent Unit (Cho et al., 2014). In this scheme, a *gate map* is generated by the concatenation of the bottom-up and the (upsampled) top-down feature map passing through a $3 \times 3$ convolution (black circle with "C" and black arrows with circle). Then a tanh function is applied to the map to generate a *gate map*. The gate map then controls the amount of contextual information that can go through by a point-wise multiplication (red lines). To make the information flow more stable, we add it with bottom-up feature map (black circle with "+"). The equations are presented in Eqs. (2). Then we use this new feature representation to replace the old one and continue feedforward calculation as described in Section 3.1.

$$O_k^{(t)} = gate * \text{Upsample}(O_{h(k)}^{(t-1)}) + O_k^{(t-1)} \tag{2a}$$

$$gate = \text{Tanh}(\text{Conv}_{3\times3}(\text{Concat}(\text{Upsample}(O_{h(k)}^{(t-1)}), O_{k-1}^{(t-1)}))) \tag{2b}$$

## 3.3 LOCATION OF RECURRENT CONNECTIONS

Since there exists a gap between the semantic meanings of feature representations of bottom-up and top-down layers, we argue that recurrent connection across too many layers can do harm to the performance. Therefore, we derive three sets of connections, `conv3_2` to `conv2_2`, `conv4_2` to `conv3_3`, and `conv5_2` to `conv4_3` respectively. It is worth noting that all these connections go across pooling layers, for pooling layers can greatly enlarge the receptive field of neurons and enrich the contextual information of top-down information flow. For information flow in networks with multiple recurrent connections, take the network structure in Figure 2 as an example. The part between `conv2_2` and `conv5_2` will be unrolled for a certain number of times. To make the experiments setting consistent, we used model with two recurrent connections(loop1 + loop2) in all the tasks.

## 4 EXPERIMENTS AND ANALYSIS

We first tested the Contextual Recurrent Convolutional model on standard image classification task including CIFAR-10, CIFAR-100, ImageNet and fine-grained image classification dataset CUB-200.

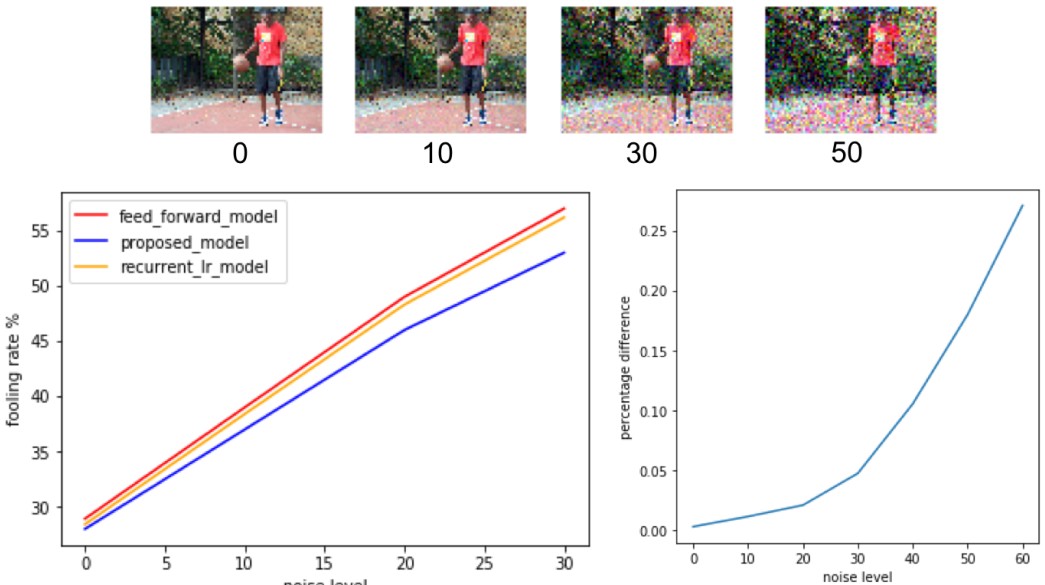

Figure 4: The example images and results of noise image classification experiment. Upper four images show an example of images with different levels of Gaussian noise added. From left to right, the standard deviations are 0, 10, 30, 50, respectively. Lower right figure shows the increased percentage of our unroll-2-times model on top-1 noise image accuracy compared with feedforward model. Lower left figure shows the adversarial attack result. The fooling rate is measured by the absolute accuracy drop when adversarial attack is performed on the model. We use standard FGSM attack on all ImageNet validation images. The blue line shows the fooling rate of our unroll-2-times model, red line shows the feed-forward model and the orange line shows the model proposed by (Li et al., 2018). As the attack gets stronger, our model shows more robustness.

| Model | CUB-200 |
|---|---|
| VGG-small | 64.88 |
| VGG-ATT (Jetley et al., 2018) | 73.19 |
| VGG-LR-2 (Li et al., 2018) | 72.99 |
| VGG-CRCN-2 | **74.90** |

| Model | Occlusion |
|---|---|
| VGG-small | 34.50 |
| VGG-ATT (Jetley et al., 2018) | 46.57 |
| VGG-LR-2 (Li et al., 2018) | 45.88 |
| VGG-CRCN-2 | **50.70** |

Table 2: Top-1 accuracy on CUB-200 datasets.  Table 3: Top-1 accuracy on Occlusion datasets.

To display the robustness of our model, we showed its performance on noise image classification, adversarial attack and occluded images. We found that our model achieved considerate performance gain compared with the standard feedforward model on all these tasks. Notice that our proposed models are based on VGG16 with 2 recurrent connection(loop1+loop2 in Figure 2) in all the tasks.

### 4.1 STANDARD IMAGE CLASSIFICATION

**CIFAR-10:** Because CIFAR-10 and CIFAR-100 datasets only contain tiny images, the receptive fields of neurons in layers beyond `conv3_2` already cover an image entirely. Although the real power of contextual modulation is hindered by this limitation, our model can still beat the baseline VGG16 network by a large margin (Second column in Table 1). Our model also compared favorably to two other recent models with recurrent connections. Again, our models showed better results.

**CIFAR-100:** Based on the assumption that contextual modulation can help layers capture more detailed information, we also tested our model on CIFAR-100 dataset, which is a 100-category version of CIFAR-10. Our model got a larger improvement compared with feedforward and other models (The third column in Table.1).

| Models / Noise Level | VGG16 | Module 1 | Module 2 | Module 3 | Proposed |
|---|---|---|---|---|---|
| 0 | 71.076 | 71.608 | 71.540 | 71.500 | **71.632** |
| 10 | 65.456 | 66.400 | 66.578 | 66.580 | **66.760** |
| 20 | 54.090 | 56.630 | 55.944 | 56.040 | **56.294** |
| 30 | 39.124 | 41.090 | 41.800 | 41.520 | **42.104** |
| 40 | 24.068 | 26.980 | 27.634 | 26.910 | **27.766** |
| 50 | 13.072 | 15.890 | **16.458** | 15.460 | 16.310 |

Table 4: Noise image classification top-1 accuracy on different module structures. VGG16: standard feedforward model. module 1: top-down gating contextual. module 2: contextual gating contextual. module 3: contextual gating top-down and top-down gating contextual combined. Proposed: contextual gating top-down.

| Locations / Noise Level | Loop 1 | Loop 2 | Loop 3 | Loop 1+2 | Loop 2+3 | Loop 1+2+3 |
|---|---|---|---|---|---|---|
| 0 | 71.581 | 71.672 | 71.580 | 71.632 | 71.646 | **71.745** |
| 10 | 66.151 | 66.075 | 65.952 | 66.760 | 66.646 | **67.620** |
| 20 | 55.301 | 55.240 | 54.692 | 56.294 | 56.000 | **56.988** |
| 30 | 40.271 | 40.150 | 39.773 | 42.104 | 41.621 | **42.686** |
| 40 | 25.600 | 25.490 | 24.910 | 27.766 | 27.110 | **28.120** |
| 50 | 14.045 | 13.932 | 12.418 | 16.310 | 16.014 | **17.102** |

Table 5: Noise image classification top-1 accuracy on different loop locations. Loop1 corresponds to the first feedback connection in Figure 2. The same for Loop2, 3, 1+2, 2+3 and 1+2+3.

## 4.2 Noise Image Classification and Adversarial Attack

**ImageNet:** ImageNet (Krizhevsky et al., 2012) is the commonly used large-scale image classification dataset. It contains over 1 million images with 1000 categories. In this task, to test the robustness of our model, we added different levels of Gaussian noise on the 224px×224px images in the validation set and calculated the performance drop. In detail, we used the two recurrent connection model for this task(loop1+loop2 in Figure 2). Notice that all models are not trained on noise images. The result of top1 error without any noise is shown in Table 7. We found that the performance gap between our model and feedforward VGG model got larger as the noise level increased. Results are shown in Figure 4. Also, we showed the noise ImageNet top-1 accuracy of our model, (Li et al., 2018)'s model and feed-forward model in Table 8.

Additionally, we also tested adversarial attacks on our model. Figure 4 shows the results with different $L_\infty$ norm coefficient. We also found that our model had much lower fooling rates than feedforward model and (Li et al., 2018)'s model with the increasing of the norms, which successfully proved our model's robustness.

## 4.3 Fine-grained Image Classification

We argued that the contextual module can help the network to preserve more fine-grained details in feature representations, and thus we tested our model on CUB-200 fine-grained bird classification dataset (Wah et al., 2011). We used the same model as ImageNet classification task which indicates that our model contains two recurrent connection(loop1+loop2 in Figure 2). As a result, our model can outperform much better than the feed-forward VGG model(Zagoruyko & Komodakis, 2016) and other similar models with the same experimental settings. The result is shown in 2.

## 4.4 Occluded Image Task

To further prove the robust ability of our model, we tested our model on VehicleOcclusion dataset (Wang et al., 2017b), which contains 4549 training images and 4507 testing images covering six

types of vehicles, i.e., airplane, bicycle, bus, car, motorbike and train. For each test image in dataset, some randomly-positioned occluders (irrelevant to the target object) are placed onto the target object, and make sure that the occlusion ratio of the target object is constrained. One example is shown in Figure 6. In this task, we used multi-recurrent model which is similar with the model mentioned in Imagenet task. Here, we found that our model can achieve a huge improvement, which is shown in 3.

## 4.5 DISCUSSION AND ANALYSIS

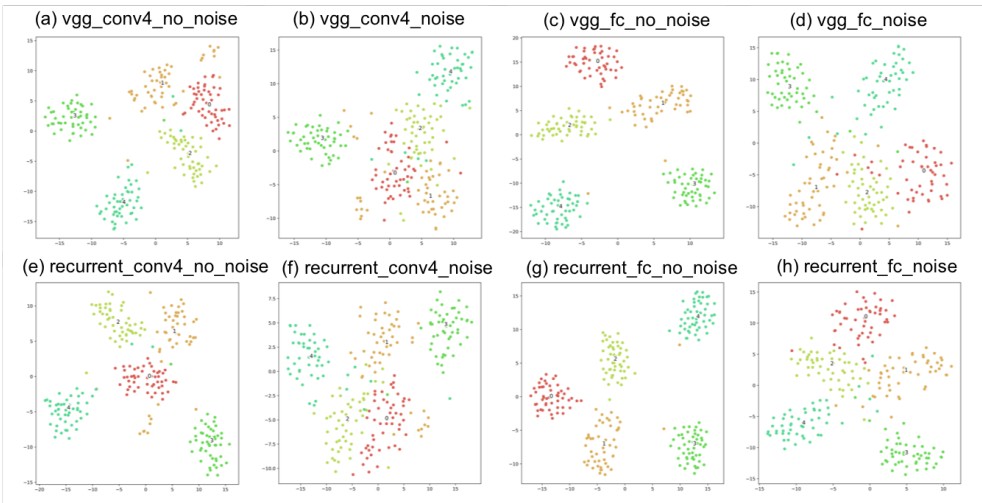

Figure 5: The results of t-SNE visualization. Upper four sub-figures shows the result of VGG16. (a) shows the result of conv4 layer without noise. (b) shows conv4 layer with noise level 30. (c) shows FC layer without noise. (d) shows FC layer with noise level 30.Lower four sub-figures shows the corresponding results of VGG-CRCN-2 model.

### 4.5.1 LOCATION OF RECURRENT CONNECTIONS

We implemented all the possible combinations of recurrent connections listed in Figure 2. We denote connection from `conv3_2` to `conv2_2`, `conv4_2` to `conv3_3`, and `conv5_2` to `conv4_3` as Loop 1, Loop 2 and Loop 3, respectively. The same naming scheme goes for Loop 1+2 and Loop 1+2+3, etc. We tested altogether 6 different models on the noise classification experiment, the settings of which were completely the same. In Table 5, by comparing the corresponding columns where one more recurrent connection is added, we can find that having more loops yields better classification accuracy and robustness, consistent with the reciprocal loops between successive layers in the hierarchical visual cortex. Especially, we can also find that the importance of Loop 1 is slightly better than Loop 2 and Loop 3, indicating the early layers may benefit more from the additional contextual information as an aid.

### 4.5.2 CONTEXTUAL MODULE STRUCTURE

In additional to the original contextual module in Figure 3, we implemented three other structures that we thought were all reasonable, so as to further study the effect and importance of top-down information and contextual modulation. Briefly, we refer Module 1 to the scheme that top-down feature map gating contextual map, Module 2 to contextual map gating contextual map itself, Module 3 to the scheme that top-down feature map gating contextual map, as well as contextual map gating top-down feature map, and afterwards the two gating results are added together. The final output of all three modules are the gating output added by bottom-up feature map. By "contextual map", we mean the concatenation of top-down and bottom-up feature map undergone a $3\times3$ convolution layer. By "gating", we mean the gated map element-wisely multiplied with the Sigmoid responses of the gate map. For formulas and further details of the three module structures, we guide readers to read the supplementary materials.

| Models
Noise Level | VGG16 | Unroll 0 | Unroll 1 | Unroll 2 | Unroll 3 | Unroll 4 |
|---|---|---|---|---|---|---|
| 0 | 71.076 | 71.018 | 71.032 | 71.221 | 71.216 | 71.612 |
| 10 | 65.456 | 66.271 | 66.368 | 66.484 | 66.481 | 66.757 |
| 20 | 54.090 | 55.810 | 55.880 | 55.938 | 55.894 | 56.291 |
| 30 | 39.124 | 41.442 | 41.492 | 41.516 | 41.551 | 42.054 |
| 40 | 24.068 | 27.588 | 28.010 | 28.044 | 28.031 | 28.102 |
| 50 | 13.072 | 15.860 | 15.941 | 15.954 | 15.982 | 16.271 |

Table 6: Noise image classification top-1 accuracy on different unrolling times of our proposed model. VGG16 means Feed-forward VGG16 model and Unroll x indicates Unroll x times during the test process.

We did the same noise image classification experiments on these different contextual modules to give a comparison. We use the Loop 1+2 model as the remaining fixed part. The performance of these modules are listed in Figure 4. The differences among these contextual modules lie in how the gate map is generated and what information is to be gated. The best model is obtained by generating the gate map from contextual map and then use it to gate top-down information. By comparing it with Module 1, we find that using only top-down information to generate the map and control total data flow is not adequate, possibly because top-down information is too abstract and coarse. By comparing the best module with Module 2, we find that only top-down information is necessary to be gated. A direct addition of bottom-up map with the output of the gate is adequate to keep all the details in lower level feature maps.

### 4.5.3 FEATURE ANALYSIS

We drew t-SNE visualization of feature representations of both final fully connected layers and layers with recurrent connections attached (e.g. conv2_2, conv3_3, conv4_3). We selected 5 out of 1000 categories from ImageNet validation set. To effectively capture the changes of feature representations of intermediate convolutional layers, we used ImageNet bounding box annotations and did an average pooling of all the feature responses corresponding to the object bounding box. By comparing the representations of both networks, we can find that the Contextual Recurrent Network is able to form a more distinct clustering than VGG16 network. Notice that we also tested the presentation when a high noise (standard deviation equal to 30) is added to the images. We can find a consistent improvement over VGG16 network in both intermediate representations and representations directly linked to the final classification task. The results are shown in Figure 5.

### 4.5.4 UNROLLING PROCESS

There is another finding that the contextual module dynamics in recurrent connections not only helps to refine the low-level feature representation during inference, it can also refine the feedforward weights, resulting in better performance in computer vision tasks even in the first iteration, acting as a regularizer. The results are shown in Table 6.

## 5 CONCLUSION

In this paper, we proposed a novel Contextual Recurrent Convolutional Network. Based on the recurrent connections between layers in the hierarchy of a feedforward deep convolutional neural network, the new network can show some robust properties in some computer vision tasks compared with its feedforward baseline. Moreover, the network shares many common properties with biological visual system. We hope this work will not only shed light on the effectiveness of recurrent connections in robust learning and general computer vision tasks, but also give people some inspirations to bridge the gap between computer vision models and real biological visual system.

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

| Models | Imagenet |
|---|---|
| VGG16 (Simonyan & Zisserman, 2014) | 71.076 |
| VGG-LR-2 (Li et al., 2018) | 71.550 |
| VGG-CRCN-2 | **71.632** |

Table 7: ImageNet classification top-1 accuracy.

## 6 SUPPLEMENTARY MATERIALS

### 6.1 DETAILS OF DIFFERENT CONTEXTUAL MODULES

We tested three other possible contextual modules in Section 4. Here are the detailed formulations of the three modules.

$$O_k^{(t)} = gate * contextual + O_k^{(t-1)} \tag{3a}$$

$$gate = \text{Tanh}(\text{Conv}_{1\times1}(\text{Upsample}(O_{h(k)}^{(t-1)}))) \tag{3b}$$

$$contextual = \text{Conv}_{3\times3}(\text{Concat}(\text{Upsample}(O_{h(k)}^{(t-1)}), O_{k-1}^{(t-1)})) \tag{3c}$$

$$O_k^{(t)} = gate * contextual + O_k^{(t-1)} \tag{4a}$$

$$gate = \text{Tanh}(\text{Conv}_{1\times1}(\text{Concat}(\text{Upsample}(O_{h(k)}^{(t-1)}), O_{k-1}^{(t-1)}))) \tag{4b}$$

$$contextual = \text{Conv}_{3\times3}(\text{Concat}(\text{Upsample}(O_{h(k)}^{(t-1)}), O_{k-1}^{(t-1)})) \tag{4c}$$

$$O_k^{(t)} = gate\_contextual * O_{k+h(k)}^{(t-1)} + gate * contextual + O_k^{(t-1)} \tag{5a}$$

$$gate = \text{Tanh}(\text{Conv}_{1\times1}(\text{Upsample}(O_{h(k)}^{(t-1)}))) \tag{5b}$$

$$contextual = \text{Conv}_{3\times3}(\text{Concat}(\text{Upsample}(O_{h(k)}^{(t-1)}), O_k^{(t-1)})) \tag{5c}$$

$$gate\_contextual = \text{Tanh}(contextual) \tag{5d}$$

In the module described by Eqs. (3), we first generated the gate by the top-down layer. Then we used the gate to control the contextual information generated by concatenating bottom-up layer and top-down layer. To stable the information flow, we added it with the bottom-up layer.

In the module described by Eqs. (4), we first generated the gate by contextual information which is the same as our proposed module. Then we used the gate to control the contextual information itself which we thought was a feasible way to store the largest information. To stable the information flow, we also added it with the bottom-up layer.

We generated two gates by both contextual information and top-down layer in the module described by Eqs. (5). Then we used the gate_contextual to control the top-down information and used the gate to control the contextual information. To stable the information flow, we also added it with the bottom-up layer.

### 6.2 IMAGE EXAMPLES OF DIFFERENT TASKS

In this section, we showed some examples of image occlusion task and adversarial noise task.

In the left of Figure 6, we showed one image occlusion example. And we showed one adversarial noise example in the right of Figure 6.

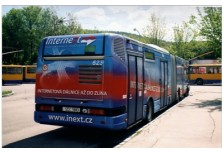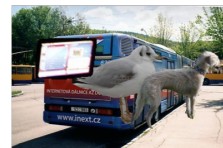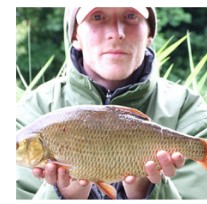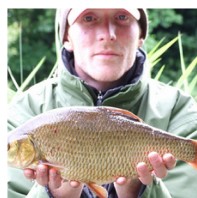

Figure 6: Examples of different task. **Left**: An example of image occlusion task. We quantified the scale of occluders in the image. **Right**: An example of Adversarial Attack noise. We can see the noise is not obvious to the human eyes but can lead a significant influence to the neural network. We used Fast Gradient Sign Non-target to generate the noise. The left is the original image and the right one is the image adding the noise.

| Models
Noise Level | VGG16 | VGG-LR-2 | VGG-CRCN-2 |
|---|---|---|---|
| 0 | 71.076 | 71.551 | **71.632** |
| 10 | 65.456 | 66.012 | **67.620** |
| 20 | 54.090 | 54.640 | **56.988** |
| 30 | 39.124 | 39.634 | **42.686** |
| 40 | 24.068 | 24.721 | **28.120** |
| 50 | 13.072 | 13.907 | **17.102** |

Table 8: Noise image classification top-1 accuracy on Imagenet.

## 6.3 IMAGENET TOP1 ACCURACY

In Table 7, we showed the Imagenet Top1 accuracy results. Notice that we did not compare our model with VGG-ATT model proposed in (Jetley et al., 2018) because their model is not reasonable on high resolution image dataset. Therefore, their model cannot extract effective attention map from the ImageNet images.

## 6.4 NOISE IMAGENET TOP1 ACCURACY

In Table 8, we showed the Imagenet Top1 accuracy results with different level of Gaussian noise. VGG16 here means the standard VGG16 model. Notice that we also compared our model with (Li et al., 2018)'s model which we name "VGG-LR-2".

