# OpenReview forum: "Contextual Recurrent Convolutional Model for Robust Visual Learning"
_ICLR.cc/2019/Conference_

### Official Review · AnonReviewer2 · 2018-11-02
**An experimental mess**

**Rating:** 4
**Confidence:** 5

**Review:**

This paper presents a novel deep learning module for recurrent processes. The general idea and motivation are generally appealing but the experimental validation is a mess. Architectures and hyper-parameters are casually changed from experiment to experiment (please refer to Do CIFAR-10 Classifiers Generalize to CIFAR-10? By Recht et al 2018 to understand why this is a serious problem.) Some key evaluations are missing (see below). Key controls are also lacking. This study is not the first to look into recurrent / feedback processes. Indeed some (but not all) prior work is cited in the introduction. Some of these should be used as baselines as opposed to just feedforward networks. TBut with all that said, even addressing these concerns would not be sufficient for this paper to pass threshold since overall the improvements are relatively modest (e.g., see Fig. 4 right panel where the improvements are a fraction of a % or left panel with a couple % fooling rates improvements) for a module that adds significant computational cost to an architecture runtime. As a side note, I would advise to tone down some of the claims such as "our network could outperform baseline feedforward networks by a large margin”...

****
Additional comments:

The experiments are all over the place. What is the SOA on CIFAR-10 and CIFAR-100? If different from VGG please provide a strong rationale for testing the circuit on VGG and not SOA. In general, the experimental validation would be much stronger if consistent improvements were shown across architectures.

Accuracy is reported for CIFAR-10 and CIFAR-100 for 1 and 2 feedback iterations and presumably with the architecture shown in Fig. 1. Then robustness to noise and adversarial attacks tested on ImageNet and with a modification of the architecture. According to the caption of Fig. 4, this is done with 5 timesteps this time! Accuracy on ImageNet needs to be reported ** especially ** if classification accuracy is not improved (as I expect).

Then experiments on fine-grained with ResNet-34! What architecture is this? Is this yet another number of loops and feedback iterations? When reporting that "Our model can get a top-1 error of 25.1, while that of the ResNet-34 model is 26.5.” Please provide published accuracy for the baseline algorithm.

For the experiment on occlusions, the authors report using “a multi-recurrent model which is similar to the model mentioned in the Imagenet task”. Sorry but this is not good enough.

Table 4 has literally no explanation. What is FF? What are unroll times?

As a side note, VGG-GAP does not seem to be defined anywhere.

When stating "We investigated VGG16 (Simonyan & Zisserman, 2014), a standard CNN that closely approximate the ventral visual hierarchical stream, and its recurrent variants for comparison.”, the authors probably meant “coarsely” not “closely".

---

> ### Author Response · Authors · 2018-11-28
> **Point-by-point address to reviewer's concerns [part 1]**
>
> Thank you for the valuable feedback and comments. Below we address your comments point by point.
>
> 1. Architectures and hyper-parameters are casually changed from experiment to experiment (please refer to Do CIFAR-10 Classifiers Generalize to CIFAR-10? By Recht et al 2018 to understand why this is a serious problem.)
>
> To address your concern, we now choose one model VGG models with loop1+loop2 (Figure 2) and test it against the other models in all the five tasks.
>
> 2. This study is not the first to look into recurrent / feedback processes. Indeed some (but not all) prior work is cited in the introduction. Some of these should be used as baselines as opposed to just feedforward networks.
>
> Yes, now we implemented Li et al’s VGG-LR (learning to rethink), and Jetley et al’s VGG-ATT (attention), and test them in addition to our model and VGG in the different tasks. (see general response). We demonstrated in all these tasks, our model provides the best performance.
>
> 3.Overall the improvements are relatively modest (e.g., see Fig. 4 right panel where the improvements are a fraction of a % or left panel with a couple % fooling rates improvements)
>
> We have corrected this mistake.
> Generally, in all tasks, our model outperformed the best of the other models by one or two percentage point. But in fine-grained recognition and noisy image recognition, our performance improvement reached 8% and 25% more than the best among the other models.
>
> 4. The experiments are all over the place. What is the SOA on CIFAR-10 and CIFAR-100? If different from VGG please provide a strong rationale for testing the circuit on VGG and not SOA. In general, the experimental validation would be much stronger if consistent improvements were shown across architectures.
>
> We now limit the comparison between one version of our model (the optimal one) against VGG and the other two VGG models with loops VGG-ATT and VGG-LR. VGG is used because it resembled the primate visual system the most in many aspects – gradual increase in receptive fields, convolution, and pooling. We consider each VGG stage, which includes multiple convolution layers followed by a pooling layer, to be equivalent to one visual area, hence, although VGG16 has 16 layers, it roughly has 6-7 stages, approximating the primate hierarchical visual system.
>
> 5. Accuracy is reported for CIFAR-10 and CIFAR-100 for 1 and 2 feedback iterations and presumably with the architecture shown in Fig. 1.
>
> Yes. The architecture used for CIFAR-10 and CIFAR-100 is VGG16 model with loop1+loop2, which is shown in Fig 1, Fig 2 and Fig 3.

---

> > ### Author Response · Authors · 2018-11-28
> > **Point-by-point address to reviewer's concerns [part 2]**
> >
> > 6. Then robustness to noise and adversarial attacks tested on ImageNet and with a modification of the architecture. According to the caption of Fig. 4, this is done with 5 timesteps this time!
> >
> > We apologize that we actually used 2 unroll times model for ImageNet classification instead of 5 unroll times. We have corrected this mistake.
> >
> > 7. Accuracy on ImageNet needs to be reported ** especially ** if classification accuracy is not improved (as I expect).
> >
> > We have reported the top-1 accuracy of ImageNet classification in Table 7 for your reference.
> >
> > 8. Then experiments on fine-grained with ResNet-34! What architecture is this? Is this yet another number of loops and feedback iterations? When reporting that "Our model can get a top-1 error of 25.1, while that of the ResNet-34 model is 26.5.” Please provide published accuracy for the baseline algorithm.
> >
> > We mentioned in the paper that “Notice that our proposed models are based on VGG16 with 2 recurrent connection (loop1+loop2 in Figure 2) in all the tasks.”
> > And we have systematically compared our model against three baseline models (VGG, VGG-ATT, VGG-LR) on fine-grained image classification dataset on Table 2.
> >
> > 9. For the experiment on occlusions, the authors report using “a multi-recurrent model which is similar to the model mentioned in the Imagenet task”. Sorry but this is not good enough.
> >
> > Now, we used the same version of our model in all tasks.
> > We have systematically compared our model against three baseline models (VGG, VGG-ATT, VGG-LR) on occlusion dataset on Table 3.
> >
> > 10. Table 4 has literally no explanation. What is FF? What are unroll times? As a side note, VGG-GAP does not seem to be defined anywhere.
> >
> > We apologize for not explaining everything clearly. FF here means feed-forward which is equal to VGG16 feed-forward model.
> >
> > 11.When stating "We investigated VGG16 (Simonyan & Zisserman, 2014), a standard CNN that closely approximate the ventral visual hierarchical stream, and its recurrent variants for comparison.”, the authors probably meant “coarsely” not “closely".
> >
> > We agree with you and have changed it to “coarsely”. But VGG is probably the best among the various deep networks to the ventral stream, in terms of the gradual growth of the receptive field size, the number of stages. A number of studies showed that V1 neurons in monkeys are closest to conv2.3 in some datasets (Kohn and Schwartz dataset) and conv3.1 of VGG (Tolias dataset, and Tang dataset).

---

### Official Review · AnonReviewer3 · 2018-11-03
**The paper proposes to add "recurrent" connections inside a convolution network with gating mechanism. The idea is not novel and the performance improvement is marginal.**

**Rating:** 3
**Confidence:** 5

**Review:**

The paper proposes to add "recurrent" connections inside a convolution network with gating mechanism. The basic idea is to have higher layers to modulate the information in the lower layers in a convolution network. The way it is done is through upsampling the features from higher layers, concatenating them with lower layers and imposing a gate to control the information flow. Experiments show that the model is achieving better accuracy, especially in the case of noisy inputs or adversarial attacks.

- I think there are lot of related literature that shares a similar motivation to the current work. Just list a few that I know of:
Ronneberger, Olaf, Philipp Fischer, and Thomas Brox. "U-net: Convolutional networks for biomedical image segmentation." International Conference on Medical image computing and computer-assisted intervention. Springer, Cham, 2015.
Lin, Tsung-Yi, et al. "Feature Pyramid Networks for Object Detection." CVPR. Vol. 1. No. 2. 2017.
Newell, Alejandro, Kaiyu Yang, and Jia Deng. "Stacked hourglass networks for human pose estimation." European Conference on Computer Vision. Springer, Cham, 2016.
Yu, Fisher, et al. "Deep layer aggregation." arXiv preprint arXiv:1707.06484 (2017).
The current work is very similar to such works, in the sense that it tries to combine the higher-level features with the lower-level features. Compared to such works, it lacks both novelty and insights about what works and why it works.

- The performance gain is pretty marginal, especially given that the proposed network has an iterative nature and can incur a lot of FLOPs. It would be great to show the FLOPs when comparing models to previous works.

- It is interesting observation that the recurrent network has a better tolerance to noise and adversarial attacks, but I am not convinced giving the sparse set of experiments done in the paper.

Overall I think the current work lacks novelty, significance and solid experiments to be accepted to ICLR.

---

> ### Author Response · Authors · 2018-11-28
> **Point-by-point address to reviewer's concerns**
>
> Thank you for the valuable feedback and comments. Below we address your comments point by point.
>
> 1. -I think there are lots of related literature that shares a similar motivation to the current work. Just list a few that I know of:
> Ronneberger, Olaf, Philipp Fischer, and Thomas Brox. "U-net: Convolutional networks for biomedical image segmentation." International Conference on Medical image computing and computer-assisted intervention. Springer, Cham, 2015.
> Lin, Tsung-Yi, et al. "Feature Pyramid Networks for Object Detection." CVPR. Vol. 1. No. 2. 2017.
> Newell, Alejandro, Kaiyu Yang, and Jia Deng. "Stacked hourglass networks for human pose estimation." European Conference on Computer Vision. Springer, Cham, 2016.
> Yu, Fisher, et al. "Deep layer aggregation." arXiv preprint arXiv:1707.06484 (2017). The current work is very similar to such works, in the sense that it tries to combine the higher-level features with the lower-level features. Compared to such works, it lacks both novelty and insights about what works and why it works.
>
> **Novelty**: Yes. Indeed, it has been long recognized that deep neural network is missing a key feature in cortical processing and there are a variety of feedback mechanisms used in the literature (1) concatenation of feedforward and feedback responses (e.g in U-net and others listed by reviewer 2); (2) unfolding feedback into a feedforward networks (U-net, autoencoder); (3) gating, or dot product (multiplication) as in Attentional network which use high-level semantic information to gate object segmentation and localization at lower layer; (4)  recurrent with gating as in LSTM and GRU, for keeping and using internal state and memory to gate sequence processing; (5) scaling, addition and subtraction typical in neuroscience. (6) Gated Boltzmann machine and transformer network. (7) capsule network. What we investigated, however, are neuroscience motivated questions: Why loops are predominantly between adjacent visual areas? What kind of contextual information would be useful and for what tasks? What is contextual modulation? These questions have not been answered by those technology-driven innovations.
> The gating circuit we proposed are also neurally motivated, and designed to answer neuroscience question. For that reasons, we might also be the first to introduce loops between convolution layers and successfully train on real complicated image dataset(ImageNet) at the same time.
>
> **Insights**: Our work did provide a number of insights discussed in our general comments. We show loops are good, tight loops are better, more loops are better, top-down gating is important, context information should include both top-down and bottom information, and feedback improves feedforward connections.
>
> 2. - The performance gain is pretty marginal, especially given that the proposed network has an iterative nature and can incur a lot of FLOPs. It would be great to show the FLOPs when comparing models to previous works.
>
> The performance improvement is a means rather than an end to our research. We use it to gauge the importance of certain designs, to assess the importance of feedback in a certain task. So we are not particularly concerning with the number of FLOPs. Our primary motivation is for understanding the computational logic of brain structures, developing a new deep learning module for technology is secondary. We are afraid that we have not made these objectives clear and the paper was evaluated as a technology paper rather than a brain science paper.
>
> In the earlier submission, we mainly provided results on the noisy image object recognition, though we mentioned results in adversary attacks and occlusion and fine-grained recognition, and we only compared our model against VGG. Now, we have implemented the other VGG-with-loops models such as VGG-ATT and VGG-LR and tested them as well. These models have similar FLOP to our model. We demonstrated our model is better.
>
> 3.- It is interesting observation that the recurrent network has a better tolerance to noise and adversarial attacks, but I am not convinced giving the sparse set of experiments done in the paper. Overall I think the current work lacks novelty, significance and solid experiments to be accepted to ICLR.
>
> Based on your review that our performance gain is limited and the experiments are sparse, to address your concerns, we systematically compared our model against other baseline models in five different tasks (recognition in Cifar10 & 100 datasets, fine-grained image classification dataset CUB-200, occlusion dataset, and ImageNet with different level of noise, adversarial attack) and provided five additional tables (Table 1,2,3,7,8) and a figure(Figure 4) for systematic comparison. We systematically compared feedforward VGG, VGG-ATT and VGG-LR, and we demonstrated in all these tasks, our model provided the best performance, thus demonstrating the computational advantages of some of the neural constraints and design.

---

### Official Review · AnonReviewer1 · 2018-11-04
**an ok paper but not good enough**

**Rating:** 4
**Confidence:** 4

**Review:**

This paper introduces feedback connection to enhance feature learning through incorporating context information.

A similar idea has been explored by (Li, et al. 2018). Compared with that work, the novelty of this work is weaken and seems limited. The difference from Li is not very clear. The authors need to give more discussion. Furthermore, experimental comparison with Li et al. 2018 is also necessary.

The performance gain is limited. The authors mainly evaluate their method for noisy image classification. Such application is very narrow and deviates a bit from realistic scenarios.

---

> ### Author Response · Authors · 2018-11-28
> **Point-by-point address to reviewer's concerns**
>
> Thank you for the valuable feedback and comments. Below we address your comments point by point.
>
> 1.A similar idea has been explored by (Li, et al. 2018). Compared with that work, the novelty of this work is weaken and seems limited. The difference from Li is not very clear. The authors need to give more discussion.
>
> Novelty: Our proposed circuit has two novel neurally inspired novel recurrent circuit design: tight loops between adjacent visual areas, and contextual modulation.
> Li et al. proposed a recurrent module bringing semantic information from the FC layer to help a lower layer perform object detection and segmentation. However, in the visual cortex, most of the loops are between adjacent areas, between V1 and V2, between V2 and V4. Such **tight loop** has not been investigated earlier, hence the problem is **novel**.  Besides we also investigated different design of contextual modulation. (see general comments above).
>
> 2.Furthermore, experimental comparison with Li et al. 2018 is also necessary.
>
> We implemented Li’s VGG-LR (Li, et al. 2018) for comparison in different tasks. Both models were unrolled 2 times during the training. In ImageNet classification task, our work shows a slightly improvement from 71.55% to 71.632% on top1 accuracy(Table 7).  Our model showed much stronger robustness against noise corruption in recognition tasks (Figure 4).  In addition, our model also outperformed VGG-LR (as well as VGG-ATT) in CIFAR 10 & 100 datasets recognition, fine-grained image classification(CUB-200) dataset as well as recognition under occlusion (Alan Yuille’s dataset (Table 1, 2, 3).
>
> 3. The performance gain is limited. The authors mainly evaluate their method for noisy image classification. Such application is very narrow and deviates a bit from realistic scenarios.
>
> We now tested our model against the other models (VGG, VGG-ATT, VGG-LR) in 5 different tasks and in different datasets. We have added five additional tables and a figure (one for each task) to document the model’s performance. We found our design outperformed in every task, with 25% improvement relative to the other baseline model for noisy image recognition, and 8% improvement relative to others in fine-grained recognition. Whether the gain is small or large is in the eye of the beholders, but our work demonstrated some computational benefits of recurrent connections, particularly for some relevant tasks.

---

### Author Response · Authors · 2018-11-28
**General Responses to the reviewers and the program committee**

We thank reviewers for their critique and valuable suggestions. The critiques of our work center on four issues: (1) *novelty* -- a number of deep networks inspired by recurrent feedback connections have already been developed such as U-net by unrolling feedback into feedforward networks with considerable performance increase, and there are also recent works of adding loops and local circuits (Li et al. 2018, Jetley et al. 2018, Nayebi et al. 2018) to improve performance; (2) *systematic comparison* is lacking across multiple tasks and against benchmark models, particularly VGG with loops; (3) *significance* -- improvement in performance seems marginal and unimpressive; (4) *insights* -- only show performance increase without explaining what work and why it works.

We agreed and disagreed but blamed ourselves for not communicating our contributions clearly. Here is our general rebuttal, followed by a point-by-point response to individual reviewers’ comments.

*Novelty*:
While recent deep learning works have added loops to the feedforward networks (VGG-ATT (attention, Jetley), VGG-LR (learning to rethink – Li), and VGG-cortex (Nayebi)), their loops jump across many layers (many stages), typically bringing semantic information from the FC layer to help lower layers to do object detection and segmentation. In the visual cortex, most of the loops are between adjacent areas, between V1 and V2, between V2 and V4, between V4 and IT. The *computational advantages of these tight loops have never been demonstrated*. Hence, our investigation of *tight loops is novel*.  Note, we consider each visual area corresponds to a set of convolution layers ending with a pooling layer, thus loops between adjacent visual areas are modeled by loops between the last convolution layer before adjacent pooling layer. Another novel contribution is our systematic exploration of various *mechanisms of gated contextual modulation* in each loop to see what top-down and bottom-up (horizontal) contextual information are most useful, and whether gating is critical. Our module design involving top-down signals and *contextual modulation* signals (from the current layer and higher layer) with multiplicative interaction is novel. Even though concatenation, convolution and multiplication are standard operations, how to put them together is important.

*Systematic evaluation*:
It is true that the evaluation in our original submission is a bit haphazard. Now, we reorganized our presentation and added additional comparison tests against VGG models with loops according to the reviewers’ advice. We implemented VGG-ATT and VGG-LR because codes are not available but checked our implementation can reproduce their original reasons. Overall, we tested these two benchmark VGG-loop models and many versions of our VGG loop models in five tasks (ImageNet recognition (Table 7) , CIFAR10 and CIFAR100 recognition (Table 1), fine-grained recognition (Table 2), Noisy ImageNet recognition (Table 8 & Figure 4), adversary attack (Figure 4), recognition under occlusion (Table 3). In all tasks, our tight loop with top-down gating contextual modulation mechanisms have produced some improvement over VGG, VGG-ATT and VGG-LR models, but the most dramatic improvement was in fine-grained recognition (by 8%) and noisy ImageNet recognition (by 25% at high noise level).

*Significance*:
We believe our work are significant in several aspects. The usefulness of feedback in deep learning, and particularly that of tight loops between cortical areas, has been elusive, and not demonstrated. Thus, *identifying and demonstrating what tasks* feedback and contextual modulation can be useful is important. We have demonstrated robustness against noise and fine-grained recognition benefit significantly from feedback and contextual modulation. Scientifically, we investigated the computational benefits of contextual modulation, which involve feedback and integration of horizontal information. Figuring what works best (top-down signal and top-down and horizontal context gating each other) provides potential insight not just for future module design but also to understanding neural mechanisms. Maybe the significance lies more on providing *insights in neuroscience* than some drastic improvement in technology.

*Insights*:
Scientifically, our work provides insights to at least WHAT work, though we are still investigating WHY it works. Here is a list of interesting insights.
1.	Loops are helpful. The tighter loop is better than the wider loop in some tasks.
2.	More loops work better.
3.	Top-down gating signal and contextual information (from top-down and bottom-up horizontal) are both important.
4.	Using contextual modulation with gating work,  using contextual modulation to scale or add or subtract does not.
5.	Having recurrent feedback with contextual modulation fundamentally change the feedforward representation.

---

### Meta-Review · Area_Chair1 · 2018-12-10
**metareview: limited novelty, unconvincing experiments**

**Confidence:** 5
**Recommendation:** Reject

**Metareview:**

This paper explores the addition of feedback connections to popular CNN architectures. All three reviewers suggest rejecting the paper, pointing to limited novelty with respect to other recent publications, and unconvincing experiments. The AC agrees with the reviewers.